# Deep Learning for Live Cell Shape Detection and Automated AFM Navigation

**DOI:** 10.3390/bioengineering9100522

**Published:** 2022-10-05

**Authors:** Jaydeep Rade, Juntao Zhang, Soumik Sarkar, Adarsh Krishnamurthy, Juan Ren, Anwesha Sarkar

**Affiliations:** 1Electrical and Computer Engineering, Iowa State University, Ames, IA 50011, USA; 2Mechanical Engineering, Iowa State University, Ames, IA 50011, USA

**Keywords:** atomic force microscope, deep learning, vision-based navigation, object detection, YOLOv3

## Abstract

Atomic force microscopy (AFM) provides a platform for high-resolution topographical imaging and the mechanical characterization of a wide range of samples, including live cells, proteins, and other biomolecules. AFM is also instrumental for measuring interaction forces and binding kinetics for protein–protein or receptor–ligand interactions on live cells at a single-molecule level. However, performing force measurements and high-resolution imaging with AFM and data analytics are time-consuming and require special skill sets and continuous human supervision. Recently, researchers have explored the applications of artificial intelligence (AI) and deep learning (DL) in the bioimaging field. However, the applications of AI to AFM operations for live-cell characterization are little-known. In this work, we implemented a DL framework to perform automatic sample selection based on the cell shape for AFM probe navigation during AFM biomechanical mapping. We also established a closed-loop scanner trajectory control for measuring multiple cell samples at high speed for automated navigation. With this, we achieved a 60× speed-up in AFM navigation and reduced the time involved in searching for the particular cell shape in a large sample. Our innovation directly applies to many bio-AFM applications with AI-guided intelligent automation through image data analysis together with smart navigation.

## 1. Introduction

Atomic force microscopy (AFM), a key member of the “scanning probe microscopy” family, offers an excellent platform for performing high-resolution imaging and the mechanical characterization of organic and inorganic samples in ambient air or liquid. Atomic force microscopy is crucial for studying soft biological samples such as proteins, DNA, RNA, and live cells. It also plays a major role in measuring important parameters of binding kinetics such as the binding probability, the most probable rupture force, and the association and dissociation constants for ligand–receptor or protein–protein interactions at the single-molecule level on live cells. Cellular fate such as cell division, proliferation, differentiation, and migration is controlled by the molecular and mechanical cues of the cellular environment [1,2,3,4]. Various mechanical stimuli altered by biochemical signals are governed by cellular interaction with their microenvironment. These interactions are responsible for turning benign cancer cells into malignant [5] cancer cells, changing their stiffness, viscoelastic properties, and shape. Intracellular mechanics and the cellular interaction with the extracellular matrix (ECM) and its environment regulate cell shape [6,7,8,9,10].

Cellular morphology is regulated by maintaining a balance between extrinsic and intrinsic forces. Drugs and other stress-causing factors such as drugs alter the mechanical properties of cells in seconds to minutes, and biochemical changes in cells are manifested in hours to days. Mechanical properties can serve as early biomarkers of biochemical changes. To obtain a statistically significant data set, measuring the mechanical properties of cells of different cell shapes is important. AFM [11,12,13,14,15,16] is a noninvasive, unique platform for generating three-dimensional surface profiles and for the mechanical characterization of hard as well as soft biological samples such as RNA, DNA, protein, and live cells in its physiological environment without compromising the integrity of the sample. Single molecule interaction forces and binding kinetics for ligand–receptor interactions on live cells are also measured using AFM [17]. Prolonged and expensive sample preparation protocols involving freezing, drying, or metal coating are not used in AFM.

However, the present-day biomechanical studies performed using AFM are low-throughput and highly time-consuming because of the lack of proper tools for automatic sample selection, detection, and AFM navigation [18]. Experimentalists manually engage the AFM [19] cantilever tip on a particular live cell to measure nanomechanical properties [20] and then retract the probe to move to another new live cell for measurement. In this study, this significant limitation is addressed by proposing an artificial intelligence (AI)-enabled AFM operation framework to accelerate measurement throughput and save a significant amount of expert effort and time.

In this paper, we leverage the deep learning (DL)-based object detection and localization techniques for automating the selection of the cell locations based on the cell shape chosen by the experimentalist to analyze AFM phase-contrast images. In experimental microscopy, this concept will help future researchers to accelerate the process significantly to conduct more experiments with lower expert efforts. To automate the navigation of the AFM probe to evaluate the cells of different shapes, we train the deep neural network on an extensive dataset of phase-contrast images containing different types of cell shapes annotated by an expert. Specifically, we focus on labeling and detecting three different cell shapes: round, polygonal, and spindle. Apart from the framework and algorithms, our annotated data will be publicly available, benefiting future research. Our proposed framework will help to interpret the interrelationship between the behavior and the morphology of the live cells in accordance with the statistically significant data set of optical images and AFM biomechanical measurements of cells. Specifically, our key contributions in this paper are:A high-speed DL-based automation to accelerate the AFM probe navigation when performing biomechanical measurements on cells with the desired shape.A transfer learning approach to adapt the cell shape detection model to low-quality images captured by the AFM stage navigation camera with limited training data.A closed-loop scanner trajectory control setup ensuring the accuracy and precision of the AFM probe navigation for biomechanical quantification.A nanomechanical property characterization for representative cell shapes using the proposed framework.

The paper is organized into four sections, including this one. Next, in the “Materials and Methods” section, we discuss cell sample preparation techniques for AFM experiments, the deep-learning-based cell shape detection processes, and the closed-loop navigation strategies for the AFM probe (tip). In the “Results and Discussion” section, we summarize the results of the cell shape detection, tip navigation, and nanomechanical property characterization. Finally, we conclude our work with some future research directions.

## 2. Materials and Methods

In this section, we discuss in detail the three major parts of our work: (i) live-cell sample preparation, (ii) cell shape detection, and (iii) closed-loop navigation. We start with the live-cell sample preparation which was used to collect data for training our DL models and validate the overall framework. DL-based cell shape detection automates the AFM probe navigation, for which it is necessary to detect the presence of the cells and their shapes in a given image. Additionally, we are interested in obtaining the coordinates of the detected cell shapes, which can be further used to generate the trajectory of the AFM probe from its current location. We feed this information to the closed-loop navigation framework for guiding the AFM probe navigation.

### 2.1. Live Cell Sample Preparation for AFM Experiments

We used an NIH-3T3 cell line (CRL-1658, ATCC) to generate phase contrast images for annotation and investigate the nanomechanical properties of live cells of different shapes in this study. These cells were derived from a mouse embryo. We maintained these cells in 25 cm_2_ cell culture flasks and passaged these cells into new culture flasks and AFM-compatible dishes every 72 h for the experiments. We detached the cells from the cell culture flasks using 0.25% trypsin-EDTA, phenol red solution (catalog number: 25200056, ThermoFischer Scientific). We disposed the cell medium in the flask and washed the cells using 1 mL of a warm Trypsin-EDTA (0.25%), phenol red solution. We added 2 mL of the same solution to the flask. We incubated the flask at 37 °C for 3 min in an incubator (with 5% CO_2_ and appropriate humidity). We added 4 mL of warm DMEM complete medium to the culture flask afterward. We refer to serum-free Dulbecco’s Modified Eagle Medium (DMEM) supplemented with L-glutamine, 4.5 g/L of glucose, sodium pyruvate, 10% calf bovine serum, and 1% PS as the complete medium here. We mixed the entire cell solution thoroughly inside the flask to dislodge most of the cells adhered to the bottom of the flask. In the meantime, we added 5 mL of warm complete medium to a new culture flask and 2.5 mL of warm complete medium to each AFM-compatible dish. We dispersed 1.5 mL of the mixed cell solution in the new flask and 400 μL of the cell solution in the AFM compatible dishes. We placed the new flask and AFM-compatible dishes inside the incubator until the measurement. To explore the effects of cellular shapes on their nanomechanical properties, we took out AFM-compatible 50 mm glass-bottom dishes plated with NIH-3T3 cells from the incubator and secured the dish to the Bioscope Resolve AFM [21,22] stage (base plate) using a vacuum pump.

### 2.2. Cell Shape Detection from Microscopic Images

To detect the cell shapes, we extended the use of a state-of-the-art deep neural network developed for object detection tasks [23]. In this work, we were interested in detecting the following shapes: (i) round, (ii) spindle, and (iii) polygonal. We differentiated the spindle shape from the polygonal shape if the shape had two narrow ends. With this DL-based approach, we detected and localized cells with particular shapes. In Figure 1, we summarize the pipeline for the cell shape detection, which includes data collection and annotation, then data augmentation, training of the deep neural network, and finally outputting the detected cell shapes and their location in the image.

Cell shape detection–localization was performed using a real-time object detection framework called the *You Only Look Once* (YOLOv3) [24] algorithm. It involved mainly two tasks: identifying the object and classifying it to one of the class labels and predicting its location in an image under consideration. The advantage of using YOLOv3 is that it accomplishes both the tasks using only a single deep convolutional neural network (CNN) and a single forward pass of an input image.

The YOLOv3 processes the input image and divides it in a grid of size S×S. Using bounding boxes priors called anchor boxes, each grid cell predicts the *B* number of boxes and is expected to predict the objects whose centers lie in that grid cell. For each bounding box prediction, YOLOv3 also predicts the confidence score or objectness score using a logistic regression, which measures the probability that the object is present in the bounding box and how accurate the bounding box is. The ground truth value of the confidence score should be 1 if the bounding box prior overlaps with the ground truth box the most among other box priors and 0 otherwise. The overlap between two boxes is calculated using the intersection over union (IoU), which lies in the range [0.0, 1.0], with 1.0 for full overlap. During the inference, it predicts the IoU as a confidence score.

YOLOv3 predicts the four parameters per bounding box for its location and these are $x_{t}$, $y_{t}$, $w_{t}$, and $h_{t}$. These parameters are relative to the grid cell location. With these four parameters, YOLOv3 finds the location of the predicted bounding box by computing the coordinates of the left corner ($bb_{x}$, $bb_{y}$) and the width ($bb_{w}$) and height ($bb_{h}$) of the bounding box using the following equations:(1)$$\begin{array}{cc} \; & bb_{x}=\sigma \left(x_{t}\right)+off_{x}\\ \; & bb_{y}=\sigma \left(y_{t}\right)+off_{y}\\ \; & bb_{w}=anc_{w}*e^{w_{t}}\\ \; & bb_{h}=anc_{h}*e^{h_{t}} \end{array}$$
where ($off_{x}$, $off_{y}$) is the offset of the grid cell from the top-left corner of the image and $anc_{w}$ and $anc_{h}$ denote the width and height of the anchor box.

In addition to the bounding box location, the network also classifies the detected object into one of the class labels among the *C* number of classes in the given data set. In total, the network predicts 5 quantities per bounding box, including the location information and confidence score. Finally, the output predicted by the YOLOv3 has the shape of S×S×(B∗(5+C)).

Similar to feature pyramid networks [25], the YOLOv3 makes the predictions across three different scales obtained by downscaling the input image by the factors of 32, 16, and 8. It uses a Darknet-53 architecture (explained in the next paragraph) as a base feature extractor, combined with numerous convolutional layers to obtain the features at three different scales.

Darknet-53 is a significantly larger and enhanced version of Darknet-19, a feature extractor proposed as a backbone model in YOLO(v2) [26], by adapting the residual network [27] concept. Darknet-53 has 53 convolutional layers as opposed to having 19 layers as in Darknet-19. This backbone model is constructed by stacking multiple convolutional and residual blocks, as illustrated in Figure 2. Each convolutional block is a sequence of a convolutional layer with a filter size of 3×3 and stride of 2, batch normalization, and a leaky-ReLU. To downscale the input image or input feature map, a convolution layer with a stride of 2 is used to minimize the information loss relative to the use of pooling operations such as max-pool [27,28]. Batch normalization is used to speed up the convergence during training and to regularize the model [29]. Each residual block is a stack of two convolutional blocks with a filter size of 1×1 and 3×3, respectively, with the stride of 1. At the end of the residual block, an identity connection is implemented by adding the input of the residual block to the second convolutional block with the shortcut connection [27]. The input image is processed and downscaled by a factor of 32 through these stacks of convolutional and residual blocks to obtain the feature maps as the output of Darknet-53.

The predictions at the first scale are obtained by processing the output of the base feature extractor, Darknet-53, by adding a few convolutional layers, where the output has the shape downscaled by a factor of 32. The feature map obtained by the layer, which precedes two layers, is upsampled by 2 and concatenated with the feature map of the same resolution from Darknet-53. This maintains the low-level features from Darknet-53 and allows one to learn meaningful semantic details. Similar to the first scale prediction, the second scale prediction is obtained by feeding the upsampled feature map to several convolutional layers. The third scale prediction involves the same operations—upsampling and concatenating the feature map from Darknet-53—and passing through convolutional layers. This process is described in Figure 3. The bounding box coordinates and confidence scores were processed through the sigmoid function to obtain values in the range of 0 and 1. We were interested in multiclass classification as the cell cannot simultaneously have more than one shape, and we used the sigmoid function to output the probabilities for each cell shape. The one cell shape with the highest probability was considered the predicted cell shape. Actual coordinates of the predicted bounding box in an image were calculated using Equation (Equation 1). To select the most significant and accurate bounding box among the thousands of bounding box predictions, YOLOv3 uses a nonmax suppression technique. It removes the duplicate boxes predicted based on the confidence score and the IoU score between the predicted and the actual bounding box.

In summary, YOLOv3 can predict objects of various sizes by making multiscale predictions. It uses Darknet-53 as the backbone feature extractor, whose output is processed further to perform the predictions at three different scales. YOLOv3 directly output the coordinates of the bounding box surrounding the cell and its classified shape.

#### 2.2.1. Dataset

Training the supervised deep learning (DL) model described above required the dataset be annotated with the cell shapes in each image. We collected several microscopic images using the camera system available on the AFM platform. There were two cameras—one at the bottom and the other at the top of the AFM stage. We captured images using both cameras. The advantage of the top camera was that it could capture the AFM cantilever probe in the image, but it output a low-quality image. On the other hand, the bottom camera could capture relatively high-quality images but did not capture the AFM cantilever probe. The cantilever probe being visible in the image made it easier for navigation purposes and other experimental analyses. The resolution of the images captured by the top camera was 640 × 480, and for the bottom camera, it was 1388 × 1040. The bottom camera could capture images at different zoom levels, 10×, 20×, and 40× by varying the optical zoom level. Overall, we captured 221 images, of which 114 were captured using the bottom camera, and the remaining 107 were captured using the top camera.

Once the data were collected, the expert could annotate the cell shape by drawing the bounding box around it and labeling it with an accurate shape. For the data annotation, we used a tool available online: *Labelbox* (https://labelbox.com/) (accessed on 20 January 2021). Collecting these images was time-consuming and tedious as the user had to manually scan the cell samples and capture the images. In addition, performing the annotations, especially on low-quality images, was a painstaking task, leading to a smaller dataset with fewer annotated images. To address this challenge, we implemented data augmentation techniques on the fly (during training), which involved rotating the original images by 90° clockwise or counter-clockwise, by 180°, flipping them upside down, and by left-right mirroring. This enhanced the original dataset with more data samples with different orientations, which further made the DL network robust to the variety of cell shape orientations encountered during inference.

#### 2.2.2. Training with Transfer Learning

The training of the YOLOv3 network was performed in two stages. As mentioned, we collected images from the top and bottom cameras of the AFM platform, producing relatively low- and high-quality images, respectively. Ideally, we needed the top camera view for the ease of navigation. However, if we performed the training using only the low-quality images the top camera produced, the network performed poorly (we discuss this more in the Results Section). Note, as discussed above, we also struggled to generate a large number of annotated data samples from the top camera due to the annotation difficulty. Therefore, we started with high-quality images for training and trained for 500 epochs. We use pretrained weights trained on the COCO dataset [30] for better performance. We further fine-tuned the network for low-quality images using the transfer learning technique. Specifically, we initialized the network weights to the weights trained on high-quality images and perform training for more than 500 epochs on the whole dataset, including both high- and low-quality images.

For each stage of training, we split the dataset into training and testing sets. The 75% of the whole dataset was grouped as a training set, and the remaining 25% formed the testing set. Images were then transformed to a fixed resolution of 640 × 640 and fed as input to the network. As described in the previous section, we performed data augmentation during training. Once the image was processed through the network, we calculated the losses. Binary cross-entropy was used for the confidence score and classification of cell shapes. We used mean the squared error for the bounding box coordinates, which ensured the predicted bounding box had a significant IoU value with the ground truth box.

### 2.3. Closed-Loop Navigation of AFM Stage

The closed-loop navigation module aimed to move the AFM stage from its current location (i.e., preoperation location) to the target cell smoothly and precisely such that the probe was right above (along the vertical direction) the selected cell for AFM characterization measurements. The working process of the AFM closed-loop stage navigation process is shown in Figure 4. In our work, the conversion factor between the optical image pixels and the distance (in metric unit), i.e., the AFM coordinate system, was calibrated by using a silicon calibration sample that had 5 by 5 μm square pitches on its surface. Pixel numbers of the pitches were counted in multiple optical images of the silicon sample to calculate the pixel size in metric unit, thus obtaining the pixel-to-distance conversion factor. As the network was also trained to recognize the AFM probe, the probe location in the optical image was used as the reference point for stage navigation. With the cell shape identification and selection performed, the desired AFM stage navigation trajectory (i.e., the stage position vs. time profile) was generated using the distance (*x* and *y* distances on the horizontal plane) between the current location of the AFM probe and the preoperation location of the selected cell to be measured.

Specifically, the navigation trajectory was generated using the linear function (with a constant velocity of 100 μm/s used in this work) with parabolic blends (with a constant acceleration of ∼500 μm/s^2^). For substantial distances, closed-loop trajectory tracking control was applied to the AFM *x*- and *y*-axis stage motors. The fine-tuning of the AFM stage position or minute distance travel was achieved by applying a closed-loop trajectory tracking control to the *x*- and *y*-axis stage with piezoelectric actuators (PEAs). Specifically, the (*x*, *y*) location of the AFM sample stage was measured in real time and fed back to the closed-loop trajectory tracking control. For large range travels (i.e., coarse-stage position travel), PI controllers were applied to drive the AFM *x*- and *y*-axis stage motors. Once the selected cell position (i.e., stage position) was within the ∼5 μm/s radius range relative to the AFM tip location, the navigation actuation was switched to the stage PEAs for fine position adjustment if necessary. Note that the PEA nonlinear dynamics [31,32], especially at high-speed operation, may affect the trajectory tracking accuracy; thus, a more advanced closed-loop control approach than the PI controller, such as model predictive control (MPC) [33], could be used to control the stage PEAs. Furthermore, to ensure the positioning accuracy, nonlinear PEA dynamics models trained using ML-based approaches could be implemented [32] as well.

## 3. Results and Discussion

We first discuss the cell shape detection and localization results. We then present an example of automatic navigation of the AFM tip. Finally, based on the cell sample measurements performed by our automated framework, we present the statistical analysis of the nanomechanical properties, which demonstrates the feasibility of our approach for scientific investigations.

### 3.1. Cell Shape Detection and Localization

We conducted two experiments: (i) we trained the network on low-quality images, and (ii) we trained it on the whole dataset, including low- and high-quality images using the transfer learning technique. We observed poor performance when trained on only the low-quality images. To overcome this challenge, we implemented a transfer learning approach, as mentioned in the earlier section, where we first trained the network on high-quality images, and then we fine-tuned the learned weights by training further using low- and high-quality images. We evaluated the performance on a test set (25% of the whole dataset, 55 images), which contained both low- and high-quality images. In this section, we compare and discuss the performance of both experiments.

We performed numerous experiments with different permutations of neural network parameters such as batch size, number of epochs, optimizers (such as Adam [34], stochastic gradient descent (SGD) [35]) for backpropagation, and learning rate. In Table 1, we summarize these experiments and report the corresponding evaluation metric, the mean average precision (mAP) over all cell shapes. The best-performing model was selected, which achieved the highest mAP value. The best model was trained for 500 epochs using a batch size of 16, with an SGD optimizer to perform a gradient descent with a cosine decaying learning rate [36], where the learning rate gradually increased to 0.001 as the training progressed and gradually fell to 0.0002 over 500 epochs. We evaluated the best model with more metrics and discuss the results below.

The performance of the object detection network was evaluated on the test data set using various metrics. Combining with numerical metrics, we displayed the accuracy of the cell shape detection by visualizing the predictions obtained using YOLOv3. We used the confusion matrix (CM) to assess the classification performance. The CM specified the number and percentage of correctly (true positives, true negatives) and incorrectly (false positives, false negatives) classified class labels for bounding boxes. Values along the columns (’True’ classes) were normalized. We compared the confusion matrices (CMs) (Figure 5) for both of the experiments and observed that the network trained on low-quality images alone could only detect on average 44% of the labeled boxes correctly; on the other hand, with transfer learning, we observed on average 63% of correct classification of cell shapes (diagonals of the matrices). With the transfer learning approach, we could reduce the false negatives (bottom rows of matrices) from almost 50% to 29%. This meant that the network failure rate to detect the cell samples was lower with transfer learning. From the last columns of the CMs, we deduce that the trained network could detect the unlabeled boxes which contain the cell shape but were not annotated explicitly. Furthermore, we observed fewer misclassifications of the cell shapes (i.e., correct detection of cell but incorrect classification of its shape) after transfer learning.

Further, we calculated the mean average precision (mAP) for all cell shapes. The value of mAP falls between 0 and 100, with the higher value desirable for better performance of the classification as well as localization. The mAP for all shapes was computed using each shape’s mean of its average precision (AP). The AP value was the area under the precision–recall (PR) curve. Precision (P) specifies the number of true positives among the total number of predicted positives, whereas recall (R) states the number of true positives among the total number of actual positives. The PR curve was generated by setting a fixed IoU threshold value and then calculating the precision and recall values using a range of confidence score thresholds. We show the PR curve for each shape and the mean over all shapes in Figure 6. We computed the AP value using a IoU threshold of 0.5, which imposes the condition that the predicted box must at least overlap by 50% with the ground truth box. With the transfer learning approach, we saw a significant improvement in the mAP value from 40.3 to 66.4, an almost 65% improvement. After comparing the confusion matrix and the average precision values, we concluded that, with transfer learning, the YOLOv3 network could perform significantly better for cell shape detection–localization.

We also report the F-1 score for each cell shape and the overall F-1 score across all the cell shapes in Table 2. As we had class-imbalanced data, the number of samples representing each cell shape was not uniform, and the F-1 score determined the network’s performance better for each cell shape. The F-1 score measures the model’s classification accuracy on the test data set. It is a combined representation of the model’s precision and recall, computed as the harmonic mean of precision and recall, and the value ranges from 0.0 to 1.0; the higher, the better. We achieved a 27% better F-1 score with the transfer learning approach. Furthermore, with transfer learning, we achieved more recall value, which showed the network was less prone to predict false negatives.

Apart from the numerical metrics, we evaluated the performance of the transfer learning approach by visualizing the predicted bounding boxes and corresponding class labels and comparing them with the labeled images. We performed inference and visualized both low- and high-quality images in Figure 7 and Figure 8, respectively. The visuals are arranged so that the ground truths/targets are shown in the top row, and the predictions are in the bottom row. Visualizing the predictions demonstrates the cell shape detection performance as it can detect most labeled shapes. Additionally, it can detect many cell shapes that were not labeled in the ground truth, as we understood from the last column of the confusion matrix. We notice that the predicted (but unlabeled) shapes are almost all correctly classified.

### 3.2. Tip Navigation

We demonstrated the proposed cell shape detection-localization framework on a commercial AFM platform (BioResolve, Bruker Inc., Billerica, MA, USA) for the automatic navigation process. As shown in Figure 9, we identified cells with different shapes. In the example shown here, we moved the AFM probe to a spindle-shaped cell (e.g., right over the cell nucleus region) from its original (preoperation) location. The cell shape detection algorithm could output the coordinates of the detected shapes in an image. With this, we had the location and shape information about all present cell shapes in an image. Combining this with the navigation mechanism, we could reduce the navigation time of the AFM platform by approximately 60×. For example, manual sample identification and navigation to 10 different cell shapes takes approximately an hour. The same can be performed within a minute using the proposed framework.

### 3.3. Nanomechanical Properties

The nanomechanical properties (adhesion, Young’s modulus, deformation, and dissipation) of live cells were measured using the PeakForce QNM mode of the Bioscope Resolve AFM platform using specialized live-cell probes to prevent damage to both the AFM cantilever probe and live cells. PeakForce QNM mode (an improved version of AFM tapping mode) [37,38,39,40] can produce high-resolution images as well as produce force–distance curves at each pixel, execute all necessary calculations on the fly, and generate high-resolution maps of nanomechanical properties.

The force–distance curves at each pixel representing the variation of the force acting on the probe vs. tip–sample separating distance were fit with the Hertz model (spherical indenter) to calculate Young’s modulus values. The Hertz model parameters are given by:(2)$$\begin{array}{cc} \; & F=\frac{4}{3}\frac{E}{\left(1-\nu^{2}\right)}\sqrt{R}\delta^{\frac{3}{2}} \end{array}$$
where *F* is the indentation force, *E* is Young’s modulus, ν is the Poisson ratio, δ is the indentation, and *R* is the radius of the indenter. The same PFQNM-LC-CAL-A probe with a calibrated spring constant of 0.101 N/m was used to perform AFM experiments on all cells of different shapes (for example, round, spindle, and polygonal shapes). Uniform scanning parameters, i.e., peak force setpoint: 539.4 pN, scan rate: 0.458 Hz, peak force amplitude: 250 nm, and peak force frequency: 1 kHz, were also maintained for all the experiments mentioned in this study.

According to Sneddon’s model [41,42], the indentation force *F* acting is given by:(3)$$\begin{array}{cc} \; & F=\frac{2}{\pi }\frac{E}{\left(1-\nu^{2}\right)}\tan\left(\alpha \right)\delta^{2} \end{array}$$
where *E* stands for Young’s modulus, ν stands for Poisson’s ratio (usually in the range of 0.2–0.5), α stands for the indenter’s half angle, and δ stands for indentation. The indentation of an infinitely hard conical cantilever tip or indenter on the elastic cylinder or sample was represented by Sneddon’s model while the indentation of an infinitely hard spherical AFM probe tip or indenter on the sample or elastic cylinder was represented by Hertz’s model. The cantilever tip radius was around 65 nm. When the indentation depth was smaller than the tip radius, the Nanoscope analysis software package applied Hertz’s model. On the other hand, Sneddon’s model was used when the indentation depths were larger than the cantilever tip radius. The trend of experimentally measured values are not going to change depending on what model is being used. However, the values might differ individually (increase or decrease depending on the tip half angle chosen by the user in NanoScope Analysis software).

Figure 10a shows the AFM peak force error image of a polygonal cell (on the left) and a spindle-shaped cell (on the right). This peak force error image shows an enhanced contrast across live cells revealing structural and topographical details of the actin cytoskeleton filament network running across the NIH-3T3 cells compared to the height sensor image that gives an idea about the height of the cell. The peak force error image represents an arbitrary measure of the regulation error of the AFM feedback loop. The actin cytoskeleton network is the major force-generating structure inside the cell that is also responsible for resisting any force applied to the cells externally by ECM or neighboring cells. Figure 10b depicts five high-resolution maps of nanomechanical properties. Observing these high-resolution maps makes it evident that on top of long actin filaments, Young’s modulus is much higher (around three times), and deformation values are much lower than in other regions of the cell body. Although adhesion should be generally added to Equation (Equation 2), measuring adhesion forces in this particular study would not make much sense as the functionalization of the AFM cantilever tip was not performed to probe any specific interactions between the substrate and the AFM cantilever tip.

Figure 10b shows the zoomed-in version of five high-resolution images of nanomechanical properties (height sensor, peak force error, DMT modulus, deformation, and adhesion maps). In the DMT modulus image, the values do not correspond directly to the true DMT modulus range and are part of a color-scale optimization automatically performed by Bruker Nanoscope software for better visualization. The modulus measurement could be performed in two different ways. The PeakForce QNM mode not only yielded high-resolution images of height and peak force error but also performed a force–distance curve at each pixel and executed all the necessary calculations on the fly to produce high-resolution maps of DMT modulus, deformation, and adhesion. We could choose up to 10 random points across the cell body on the DMT modulus image (.spm file), use the roughness tool to measure the DMT modulus values and take an average value for that particular cell, repeat the same process for twenty different cells of each shape, and take the average modulus value. Another way of measuring the modulus was to use the .pfc files (an example of .pfc file for the spindle-shaped cell is attached in the supplement Figure A4 that shows a Young’s modulus value of 30.49 kPa) for the same image, choose 10 different points along the cell body, perform force–distance curve measurements, and fit those curves individually to the Hertz model to compute Young’s modulus values, and then take the average. Both approaches produced similar modulus values. The actin cytoskeleton running across the cells had higher modulus values than other sections of the cell body. So, while choosing the 10 random points, we chose some points on the actin filaments and other sections of the cell body, and then we took the average.

Figure 11a shows the AFM probe approaching the live cells (cell in focus) of different shapes (spindle, round, and polygonal) and Figure 11b represents the AFM probe that has reached the substrate (live cells plated on glass bottom dishes in this case). Nanomechanical measurements were performed on 20 cells of each different cell shape type using our proposed framework. Figure 11c,d represent the height sensor, peak force error, DMT modulus, deformation, and adhesion maps of the round- and spindle-shaped cells, respectively, marked with yellow boxes in Figure 11a.

Figure 12a,b show how the Young’s modulus and deformation values vary depending on the cellular shape. We can conclude that the nanomechanical properties change along with the cell shape variation depending on these results. Young’s modulus values were calculated at ten different locations along the cell body, and the average of these values was considered. Polygonal cells with the stronger and larger actin cytoskeletal networks demonstrated the highest Young’s modulus values (average: 41.71 ± 15.68 kPa), the round cells demonstrated the lowest Young’s modulus values (average: 24.80 ± 7.78 kPa), and the spindle-shaped cells demonstrated mid-level Young’s modulus values (average: 27.92 ± 10.24 kPa). Similarly, the polygonal-shaped cells showed the lowest deformation (average: 228.19 ± 34.01 nm), and the round-shaped cells (average: 132.23 ± 47.66 nm) showed the highest deformation values.

Choosing 10 points on the cell body (some on the cytoskeleton and some on the other parts of the cell body) was a matter of choice. As modulus measurements were performed for 20 cells of each kind (a total of 60 cells) using the same method, we assumed that choosing 10 points on 20 cells (a total of 200 points) would be statistically sound to represent the mean. As the standard deviation values for these samples were low, we thought that showing the distribution of the mean would be sufficient.

## 4. Conclusions

In this paper, we utilized a deep-learning-based object detection neural network for a smoother and faster navigation of the AFM cantilever probe to the desired cell shape. We also generated and annotated a dataset of the AFM images with different cell shapes. With a deep-learning-based methodology, we detected and localized the cell shapes in an entire sample within a few seconds, making it efficient for instant navigation. We improved the performance of the deep neural network by implementing a transfer learning approach, which helped overcome the challenge of performing shape detection using low-quality AFM stage camera images using fewer training samples. Using transfer learning, we achieved an acceptably high accuracy for recognizing the cell shape and its location. Coupling this with the navigation mechanism, we speeded up the navigation to the desired shape by approximately 60× compared to manual AFM experiments. While this study showed the feasibility of our approach, cell shape classification performance can be further improved for widespread use by the AFM community, using larger annotated data sets and leveraging robust image enhancement techniques [43]. Performance can also improve dramatically with better cameras that may be available with advanced AFM platforms.

For future work, the proposed AI-driven sample selection can be incorporated with high-speed AFM scanning control techniques, which will help to increase the efficacy of AFM biomechanical study on live cells. Furthermore, optimal trajectory generation can be developed using a cell shape detection framework and we will develop an integrated closed-loop control framework to achieve seamless coordination at high speed between the AFM stage motors control and scanner piezoactuators.

## Figures and Tables

**Figure 1 bioengineering-09-00522-f001:**
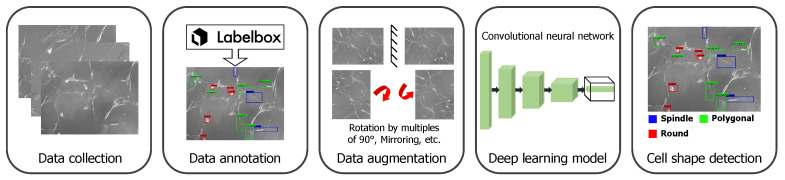
Overview of the cell shape detection pipeline. It involves data collection and augmentation and training the deep learning framework.

**Figure 2 bioengineering-09-00522-f002:**
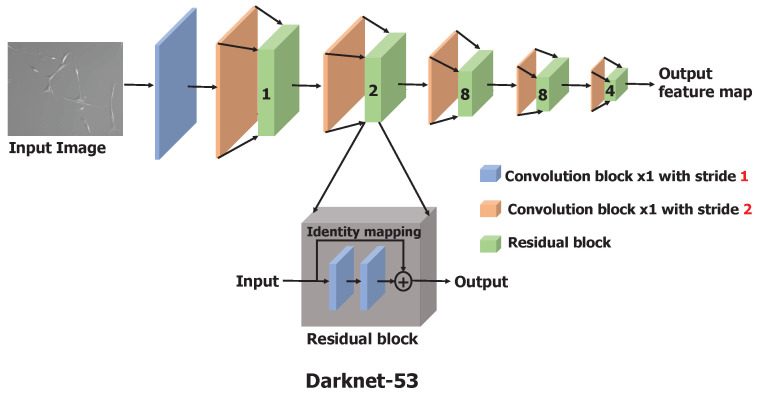
Architecture of Darknet-53, the backbone feature extractor consisting of 53 convolutional layers. Residual block composition is shown in the dark gray box. In the figure, the number shown on the green box indicates the number of residual blocks used.

**Figure 3 bioengineering-09-00522-f003:**
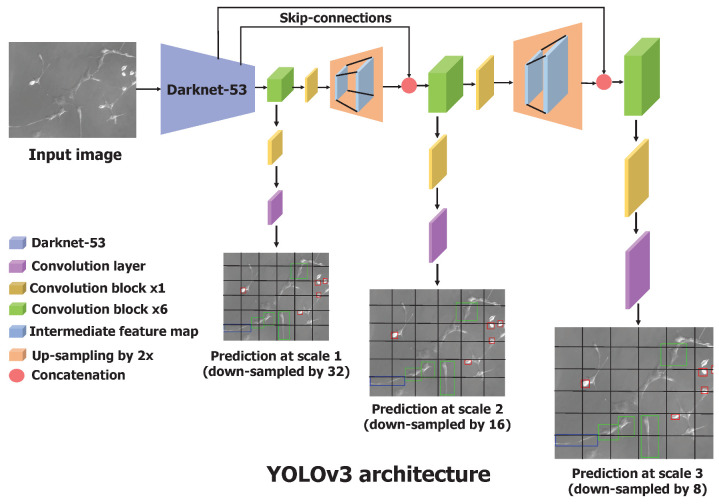
Architecture of the YOLOv3 neural network. It has a total of 106 convolutional layers. Processing the feature map obtained from Darknet-53, YOLOv3 makes predictions at three different scales and outputs the location of bounding boxes in a single forward pass. Convolutional block contain the sequence of convolutional layer, batch normalization, and leaky ReLU.

**Figure 4 bioengineering-09-00522-f004:**
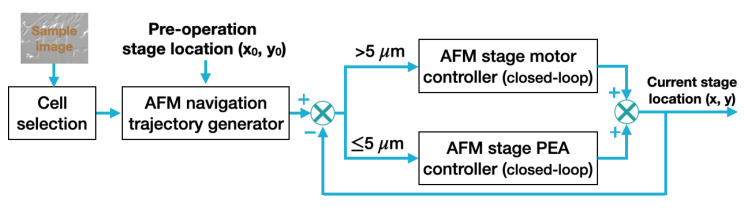
A schematic of the automatic AFM navigation closed-loop control.

**Figure 5 bioengineering-09-00522-f005:**
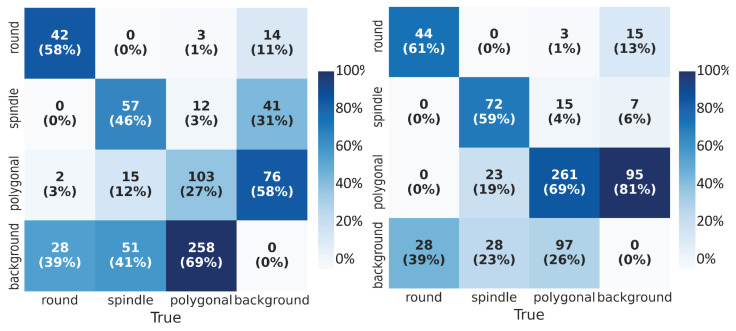
Confusion Matrix (CM): True labels on the horizontal axis and predicted labels on the vertical axis. In addition to absolute numbers, we calculate the percentage fraction along the column. (**Left**) CM for the network trained on low-quality images, (**Right**) CM for the network trained using the transfer learning technique.

**Figure 6 bioengineering-09-00522-f006:**
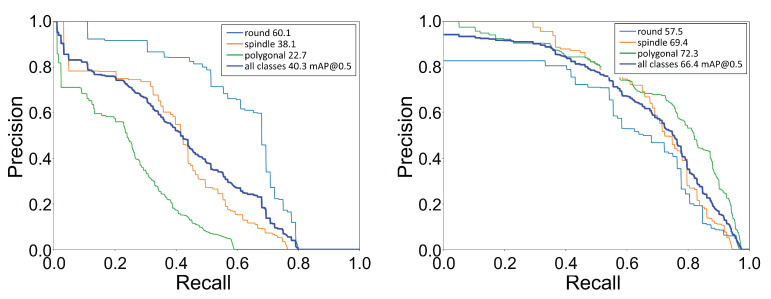
Precision-recall (PR) curve: Plot of PR curve for each cell shape with IoU threshold of 0.5. (**Left**) PR curve for the network trained on only low-quality images, mAP@0.5 = 40.3, (**Right**) PR curve for the network trained using the transfer learning technique, mAP@0.5 = 66.4.

**Figure 7 bioengineering-09-00522-f007:**
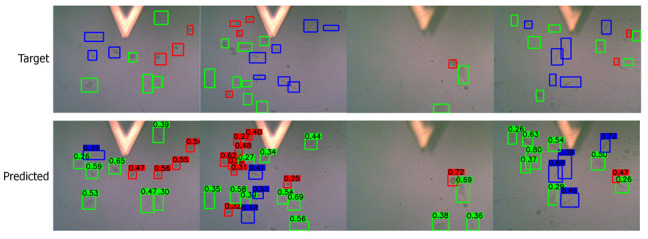
Visualizing the predictions on low-quality images. Target/ground truth images are shown in the top row and the corresponding predictions in the bottom row. The color scheme is: (i) red: round shape; (ii) blue: spindle shape; (iii) green: polygonal shape.

**Figure 8 bioengineering-09-00522-f008:**
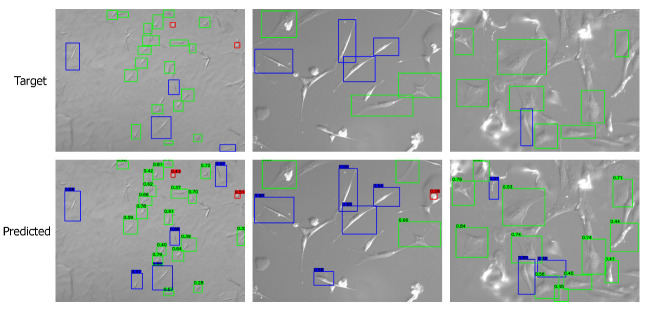
Visualizing the predictions on high-quality images. Target/ground truth images are shown in the top row and the corresponding predictions in the bottom row. The color scheme is: (i) red: round shape; (ii) blue: spindle shape; (iii) green: polygonal shape.

**Figure 9 bioengineering-09-00522-f009:**
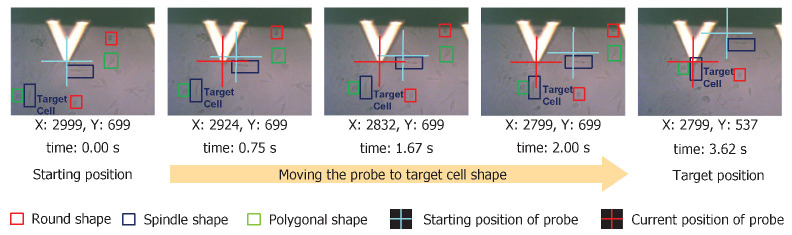
We show the sequence of images demonstrating the AFM probe navigation from starting to target position based on the cell shape identification result. We specify the (x, y) co-ordinates of the AFM probe at current location and the cumulative time to travel just below each image. Processing time was approximately 3.62 s at a navigation speed of 100 μm/s.

**Figure 10 bioengineering-09-00522-f010:**
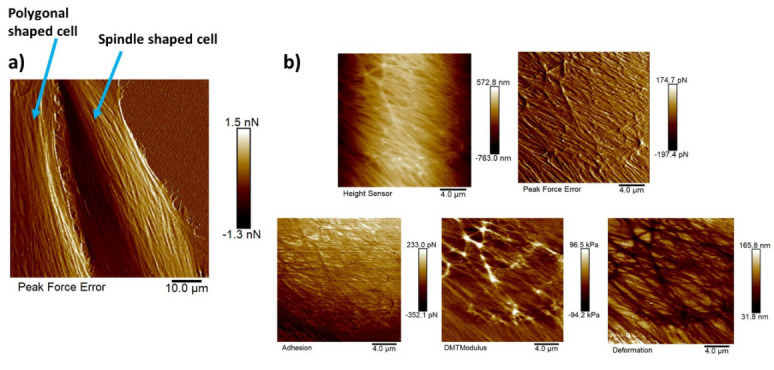
High-resolution images of NIH-3T3 cells of different shapes and high resolution maps of nanomechanical properties of a polygonal cell: (**a**) AFM peak force error image of a polygonal cell (on **left**) and a spindle-shaped cell (on **right**) revealing the actin cytoskeleton network clearly; scale bar: 10 μm. (**b**) Five high-resolution maps of nanomechanical properties (height sensor, peak force error, DMT modulus, deformation, and adhesion maps); scale bar: 4 μm.

**Figure 11 bioengineering-09-00522-f011:**
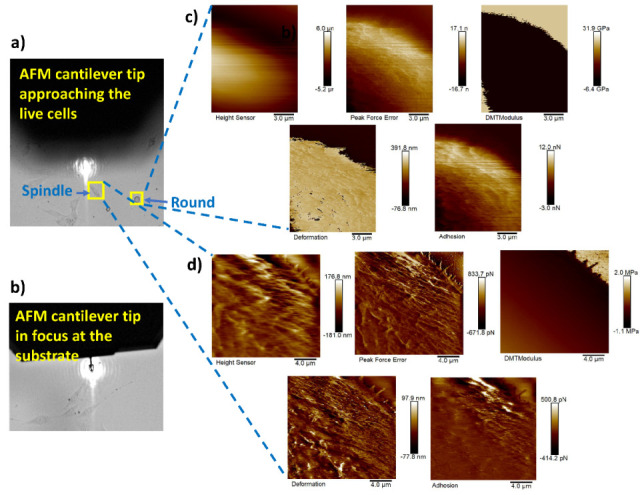
AFM tip navigation and nanomechanical measurement of round- and spindle-shaped cells: (**a**) navigation process of the AFM cantilever towards the live cells of different shapes; (**b**) cantilever tip in focus at the substrate; (**c**,**d**) height sensor, peak force error, DMT modulus, deformation, and adhesion maps of the round- and spindle-shaped cells, respectively, marked with yellow boxes in (**a**).

**Figure 12 bioengineering-09-00522-f012:**
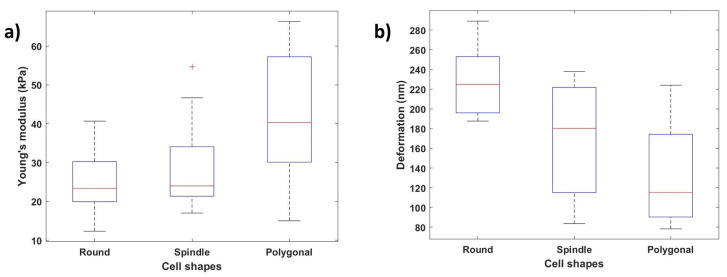
Effect of cellular shapes on Young’s modulus and deformation: (**a**) variation of Young’s modulus values with different cell shapes (round, spindle, polygonal); (**b**) variation of deformation values depending on the cellular shape.

**Table 1 bioengineering-09-00522-t001:** Comparison of models trained with different sets of neural network’s parameters. We report the mAP value over all cell shapes for each experiment and highlight the best model.

Optimizer	Batch Size	Epochs	Learning Rate	mAP
Adam	16	500	0.01	47.8
Adam	32	500	0.01	44.7
Adam	16	1000	0.01	47.3
Adam	32	500	0.01	45.9
SGD	16	500	0.01	63.8
SGD	16	1000	0.01	62.3
SGD	32	500	0.01	64.8
SGD	32	1000	0.01	64.4
SGD	16	500	0.0001	63.7
SGD	32	500	0.001	66.1
**SGD (best)**	**16**	**500**	**0.001**	**66.4**

**Table 2 bioengineering-09-00522-t002:** F-1 score for overall and each cell shape.

Experiment	Round	Spindle	Polygonal	Mean
Trained on only low-quality images	0.73	0.64	0.42	0.60
Transfer learning	0.74	**0.77**	**0.77**	**0.76**

## Data Availability

The data and the model are available upon request via this GitHub link: https://github.com/jaydeepradeJD/AFM_YOLOv3 (accessed on 22 July 2022).

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
