# Peer review of "Deep Learning for Live Cell Shape Detection and Automated AFM Navigation"

_bioengineering, 2022, doi:10.3390/bioengineering9100522_

Round 1
Reviewer 1 Report
This article implemented a deep learning framework to perform automatic sample selection and established a high speed closed-loop scanner trajectory control. In the deep learning framework, they used the YOLOv3 model with Draknet-53 as the backbone model for feature extraction. The dataset consisted of 114 high-quality images without AFM probes and 107 low-quality images with AFM probes. And the deep learning model evaluated its classification performance by mean average precision value. Their innovation will greatly facilitate the intelligence of biological AFM image data analysis. However, I think the model has some space to improve before accepting for publication.
1. The dataset consists of only 221 images. generally, deep learning requires huge datasets to guarantee its prediction performance. In this study, can the size of the dataset be increased to improve the model prediction performance? Or due to dataset limitations, other classification algorithms, such as linear models, can be used?
2. Many parameters are set in the YOLOv3 model. In this study, it is possible to improve the performance of the model for cell shape prediction by adjusting the parameters.
3. More criteria can be used to evaluate the predictive performance of the model. The AP value of 0.616 is not convincing for the reliability of the model's predictive performance.
4. In line 30: “Cell shape is regulated by cellular interaction with ECM and …”. ECM appears in the text as an abbreviation at the first time without explanation. Please add the full name.
5. In figure 1, the picture shows three cell shapes: spindle, polygonal and round. However, the spindle and the polygonal shapes in this picture is very similar to each other. So, I suggest to change to another picture which readers can tell differences of two shapes.
6. In line 121: “which includes data collection and annotation then data augmentation, training the deep…”, However there is not annotation procedure in figure 1.
7. In figure 4, AFM motor controller and AFM PEA controller are not parallel based on line 246 to 250: using motor controller for large range travels and PEA controller for small range adjustment. So, l suggest figure 4 should be re-organized.
Author Response
This article implemented a deep learning framework to perform automatic sample selection and established a high speed closed-loop scanner trajectory control. In the deep learning framework, they used the YOLOv3 model with Draknet-53 as the backbone model for feature extraction. The dataset consisted of 114 high-quality images without AFM probes and 107 low-quality images with AFM probes. And the deep learning model evaluated its classification performance by mean average precision value. Their innovation will greatly facilitate the intelligence of biological AFM image data analysis. However, I think the model has some space to improve before accepting for publication.
Response: We thank the reviewer for the accurate summary of our paper and also the positive comments. We have addressed the individual questions below.
- The dataset consists of only 221 images. generally, deep learning requires huge datasets to guarantee its prediction performance. In this study, can the size of the dataset be increased to improve the model prediction performance? Or due to dataset limitations, other classification algorithms, such as linear models, can be used?
Response: To overcome the small dataset size, we have implemented the data augmentation techniques, which help to increase the size of the dataset. Additionally, Section 2.2.1 discusses the challenges of collecting and annotating the data by hand, especially for low-quality images. We adopt a transfer learning technique to enhance the performance on low-quality images. Although, we believe that the model’s performance can be improved with a bigger dataset.
- Many parameters are set in the YOLOv3 model. In this study, it is possible to improve the performance of the model for cell shape prediction by adjusting the parameters.
Response: We summarize the experiments performed using different sets of neural network parameters in Table 1. We have experimented by changing multiple parameters, including the number of epochs used during the training, batch size and learning rate, and optimizers.
- More criteria can be used to evaluate the predictive performance of the model. The AP value of 0.616 is not convincing for the reliability of the model’s predictive performance.
Response: In the paper, we added one more metric called F-1 score in Table 1. In combination with the AP value, we also report the confusion matrix, which assesses the model’s classification performance. Adding 1 to the numerical metrics, we demonstrate the model’s performance by visualizing the low-quality and highquality images. In addition, even though the value of 0.616 seems low, given the noise level of the images, the method is comparable to manual approaches.
- In line 30: “Cell shape is regulated by cellular interaction with ECM and . . . ”. ECM appears in the text as an abbreviation at the first time without explanation. Please add the full name.
Response: We have added the full term in the text. ECM stands for extracellular matrix.
- In figure 1, the picture shows three cell shapes: spindle, polygonal and round. However, the spindle and the polygonal shapes in this picture is very similar to each other. So, I suggest to change to another picture which readers can tell differences of two shapes.
Response: We updated Figure 1, which displays different cell shapes distinctly.
- In line 121: “which includes data collection and annotation then data augmentation, training the deep. . . ”, However there is not annotation procedure in figure 1.
Response: We have updated Figure 1, showing the Data Annotation procedure.
- In figure 4, AFM motor controller and AFM PEA controller are not parallel based on line 246 to 250: using motor controller for large range travels and PEA controller for small range adjustment. So, l suggest figure 4 should be re-organized.
Response: We have updated the Figure 4 in the main paper.
Reviewer 2 Report
The authors claim to have developed a DL framework to perform automatic sample selection based on cell shape as a mechanism to reduce the amount of time required for positioning the AFM tip on top of a cell of interest. In parallel the authors also compare the elastic modulus of cells with different shapes: round, spindle and polygonal.
The research is relevant, as indeed, these types of tasks are time-consuming.
It is not clear if the authors focus only on training the network, and on showing its performance, or if this work already benefits from the algorithm. It seems it doesn’t, except for positioning the tip after recognitions as take place, is this correct?
Ideally, one would place the sample and press a button “measure cells”, the algorithm would then find the cells and take the measurements cell after cell.
Would it be possible to show a video, were initially the tip is at some random position, and no cell have been identified yet, and then we see the recognition taking place, and finally the tip moving to a specific cell, for instance the nearest identified cell? I think it would very much increase the impact of this work, and give a fair sense of how much this can facilitate life.
How did the authors arrive at the factor 60?
As for tip navigation the authors show a few images showing the tip original position and the tip’s final position. It shows that the tip can be brought from an original position to a final position close to a cell of choice. This task alone does not require any sophisticated algorithm, except that the system of coordinates measured by the optical microscope must be correlated to the system of coordinates of the AFM? How is this done? Is it already implemented in the particular AFM (bruker) , so the author didn’t have to face this? This point is not addressed. Perhaps the cantilever is used as a reference point between the two systems of coordinates? Is the network also trained to recognize the cantilever?
This point has to be addressed. How hard it would be to implement this strategy on different AFMs? Or, what are the AFM list of requirements for this to work? Can the authors address this issue?
In figure 9. besides the coordinates the authors should include the time, or else it looks like the trivial task of moving the cantilever.
In line 44 the authors mention measuring live cells in a vacuum. Is this a mistake?
On figure 10 DMT modulus, the scale bar goes from negative 94 kPa to positive 97 kPa. What is the meaning of the negative 94 kPa? Moreover, the total span is about 190 kPa, but the values presented are say, for instance, 30 pm 10 kPa, for the spindle cells? How does this happen? Also, the image reveals heterogeneity due to the actin cytoskeleton. How are the averages computed, from values coming from the actin, from the rest of the cell, or is it some blend. If it comes from some blend, how is it computed? The author mention 10 measurements, are these 10 measurements coming from random coordinates, or is there some choice?
In figure 11 one reads GPa? Shouldn’t the values be in the range of kPa?
The authors mention DMT model but equation 2 refers to Hertz model? Shouldn’t adhesion be included in equation 2?
Furthermore, if I understand well the tip radius is 65 nm, but the deformation (or indentation) is up to more than 200 nm. The model used does not take into consideration the actual tip shape at these relatively large (comparing to tip radius) indentation depths. Since each type of cell suffers different indentation what is the weight of the tip geometry in the final results? Whithout addressing this the comparison between cells can be misleading.
Other comments:
In line 65, the author state that apart from the framework ant algorithms the annotated data will be publicly available. Whereas in line 59 the authors mention “saving time to future researchers”. But without the framework other authors will have to reinvent the same thing.
The framework is only very generally addressed. It would be beneficial if the authors add more info, for instance, supplied as supplementary info, otherwise it will be hard for researchers trained in AFM (the main audience) but not in neuronal networks to replicate this.
Author Response
The authors claim to have developed a DL framework to perform automatic sample selection based on cell shape as a mechanism to reduce the amount of time required for positioning the AFM tip on top of a cell of interest. In parallel the authors also compare the elastic modulus of cells with different shapes: round, spindle and polygonal. The research is relevant, as indeed, these types of tasks are time-consuming.
Response: We thank the reviewer for the positive comments.
It is not clear if the authors focus only on training the network, and on showing its performance, or if this work already benefits from the algorithm. It seems it doesn’t, except for positioning the tip after recognitions as take place, is this correct?
Ideally, one would place the sample and press a button “measure cells”, the algorithm would then find the cells and take the measurements cell after cell.
Would it be possible to show a video, were initially the tip is at some random position, and no cell have been identified yet, and then we see the recognition taking place, and finally the tip moving to a specific cell, for instance the nearest identified cell? I think it would very much increase the impact of this work, and give a fair sense of how much this can facilitate life.
How did the authors arrive at the factor 60?
Response: We discussed the speedup factor in Section 3.2. To give more insight on it, we manually performed the cell shape detection and performed manual navigation to 10 different cell shapes which took approximately one hour. With the proposed deep-learning framework, the detection and then navigation can be performed within a minute.
As for tip navigation the authors show a few images showing the tip original position and the tip’s final position. It shows that the tip can be brought from an original position to a final position close to a cell of choice. This task alone does not require any sophisticated algorithm, except that the system of coordinates measured by the optical microscope must be correlated to the system of coordinates of the AFM? How is this done? Is it already implemented in the particular AFM (bruker) , so the author didn’t have to face this? This point is not addressed. Perhaps the cantilever is used as a reference point between the two systems of coordinates? Is the network also trained to recognize the cantilever? This point has to be addressed. How hard it would be to implement this strategy on different AFMs? Or, what are the AFM list of requirements for this to work? Can the authors address this issue?
Response: We would like to thank the reviewer for this question and the suggestions. We have added Appendix A., where we demonstrate that the network can be trained to recognize cantilever probe. As described in Figure 4, the navigation process is controlled by the closed-loop feedback mechanism as the probe location on the cell is essential to the AFM quantification result. To accomplish this, the optical images must be correlated with the AFM coordinate system, which is not implemented in the commercial AFM (Bruker) we use. In our work, the conversion factor between the optical image pixels and the distance (in metric unit), i.e., the AFM coordinate system, was calibrated by using a silicon calibration sample that has 5 by 5 μm square pitches on its surface. Pixel numbers of the pitches were counted in multiple optical images of the silicon sample to calculate the pixel size in metric unit, thus the pixel-to-distance conversion factor. As the network is also trained to recognize the AFM probe, the probe location in the optical image is used as the reference point for stage navigation. We have clarified this point in the subsection “Closed-loop Navigation of AFM Stage. This response has been added to line 240, pg. 7 in the revised manuscript. Since our approach is a generalized software package without specific requirement on the actual AFM system, it can be readily implemented on other AFM systems with the pixel-to-distance conversion factor properly calibrated.”
In figure 9. besides the coordinates the authors should include the time, or else it looks like the trivial task of moving the cantilever. The total time used was 3.5 seconds.
Response: We updated the Figure 9, which now show the cumulative timing to travel from previous to current location.
In line 44 the authors mention measuring live cells in a vacuum. Is this a mistake?
Response: We would like to thank the reviewer for his suggestion.We meant that AFM in general can perform measurements in vaccum too. Live cell experiments can only be performed in appropriate cell culture media using AFM. It has been removed from the revised manuscript.
On figure 10 DMT modulus, the scale bar goes from negative 94 kPa to positive 97 kPa. What is the meaning of the negative 94 kPa? Moreover, the total span is about 190 kPa, but the values presented are say, for instance, 30 pm 10 kPa, for the spindle cells? How does this happen? Also, the image reveals heterogeneity due to the actin cytoskeleton. How are the averages computed, from values coming from the actin, from the rest of the cell, or is it some blend. If it comes from some blend, how is it computed? The author mention 10 measurements, are these 10 measurements coming from random coordinates, or is there some choice?
Response: We would like to thank the reviewer for this useful suggestion. Figure 10 (b) shows the zoomedin version of five high-resolution images of nanomechanical properties (height sensor, peak force error, DMT modulus, deformation, and adhesion maps. In the DMT modulus image, negative 97 kPa does not really mean anything. It is just a color scale optimization automatically performed by Bruker Nanoscope software for better visualization. This color scale doesn’t represent the true DMT modulus range. The modulus measurement can be performed in two different ways. PeakForce QNM mode not only yields high-resolution images of height and peak force error but also performs a force-distance curve at each pixel, execute all the necessary calculations on the fly to produce high-resolution maps of DMT modulus, deformation, and adhesion. We can choose 10 random points across the cell body on the DMT modulus image (.spm file), use roughness tool to measure the DMT modulus values and take an average value for that particular cell and repeat the same process for twenty different cells of each shape, and take the average modulus value for plotting. Another way of measuring the modulus is to open the .pfc files (example of .pfc file for spindle shaped cell is attached in the supplementary Figure A4 that shows Young’s modulus value of 30.49 kPa) for the same image, choose 10 different points along the cell body, perform force-distance curves, and fit those curves individually to the Hertz model to yield Young’s modulus values, and then take the average. Both 3 approaches produce similar modulus values. Actin cytoskeleton running across the cells with have higher modulus values than other sections of the cell body. So, While choosing 10 different points, random points were chosen on actin filaments as well as other sections of the cell body to make sure that a blend of both values was chosen to represent the values of Young’s modulus in general and then average was taken. This response has been added in the revised manuscript (line 375, pg. 13).
In figure 11 one reads GPa? Shouldn’t the values be in the range of kPa?
Response: It is just a color scale for better visualization. The Bruker Nanoscope software optimized that image and auto-adjusted the color scale for better visualization, showing lighter color for glass substrate and a darker color for cellular surface, clearly setting them apart. The optimization scale is chosen in the GPa scale as the image shows a large glass section with Young’s modulus in the GPa range.
The authors mention DMT model but equation 2 refers to Hertz model? Shouldn’t adhesion be included in equation 2?
Response: The modulus measurement can be performed in two different ways. PeakForce QNM mode not only yields high-resolution images of height and peak force error but also performs a force-distance curve at each pixel, execute all the necessary calculations on the fly to produce high-resolution maps of DMT modulus, deformation, and adhesion. We can choose 10 random points across the cell body on the DMT modulus image (.spm file), use roughness tool to measure the DMT modulus values and take an average value for that particular cell and repeat the same process for twenty different cells of each shape, and take the average modulus value for plotting. Another way of measuring the modulus is to open the .pfc files (example of .pfc file for spindle shaped cell is attached in the supplementary Figure A4 that shows Young’s modulus value of 30.49 kPa) for the same image, choose 10 different points along the cell body, perform force-distance curves, and fit those curves individually to the Hertz model to yield Young’s modulus values, and then take the average. Both approaches produce similar modulus values. Actin cytoskeleton running across the cells with have higher modulus values than other sections of the cell body. So, While choosing 10 different points, random points were chosen on actin filaments as well as other sections of the cell body to make sure that a blend of both values was chosen to represent the values of Young’s modulus in general and then average was taken. Although adhesion should be generally added to the equation, measuring adhesion forces in this particular study won’t make much sense as functionalization of the AFM cantilever tip was not performed to probe any specific interactions between the substrate and the AFM cantilever tip. It has been clarified in line 373, pg. 13.
Furthermore, if I understand well the tip radius is 65 nm, but the deformation (or indentation) is up to more than 200 nm. The model used does not take into consideration the actual tip shape at these relatively large (comparing to tip radius) indentation depths. Since each type of cell suffers different indentation what is the
weight of the tip geometry in the final results? Without addressing this the comparison between cells can be misleading.
Response: We would like to thank the reviewer for this suggestion. Pre-calibrated live-cell probes by Bruker have been used for this study, and cantilever tip radius has already been taken into consideration for all the measurements on cells of different shapes, even at larger deformations.
In line 65, the author state that apart from the framework ant algorithms the annotated data will be publicly available. Whereas in line 59 the authors mention “saving time to future researchers”. But without the framework other authors will have to reinvent the same thing.
Response: Upon acceptance of the paper, we will make the code and dataset available on Github
The framework is only very generally addressed. It would be beneficial if the authors add more info, for instance, supplied as supplementary info, otherwise it will be hard for researchers trained in AFM (the main audience) but not in neuronal networks to replicate this.
Response: We added the Appendix B., describing the general training algorithm of neural network
Round 2
Reviewer 2 Report
The authors have addressed most of the points raised previously.
point 1: The authors mention in line 401 "So, while choosing the 10 random points, we chose some points on the actin filaments and other sections of the cell body, and then we took the average" (...) which sounds somewhat arbitrary and not too random, and yet the comparison is interesting.
I understand that this work is more focused on locating the different cells, fast and smart. Nonetheless, perhaps the authors could show the distributions of values rather than the distribution of means? Or clearly state 2 values for each type of cell: on the cytoskeleton and away?
point 2: The authors have not explained how they took into account the fact that the indentations are much larger than the tip radius. The authors mention in the reply, that they took that fact into account, but without mentioning how. Also, if I am not mistaken the tip radius is not indicated, I am inferring that it is around 65 nm. This radius is to be inserted in equation 2, but then the indentations are much larger than this radius, which implies some geometrical factor must be involved. I don't think this would change the found trend but it certainly affects the values, more so because the indentations are not identical for the different cells. In my opinion, it should at least be mentioned that due to this, the values are only approximate; and how the authors expect how it affects the results.
